# Polyamines as Quality Control Metabolites Operating at the Post-Transcriptional Level

**DOI:** 10.3390/plants8040109

**Published:** 2019-04-24

**Authors:** Laetitia Poidevin, Dilek Unal, Borja Belda-Palazón, Alejandro Ferrando

**Affiliations:** 1Instituto de Biología Molecular y Celular de Plantas, Consejo Superior de Investigaciones Científicas-Universitat Politècnica de València, 46022 Valencia, Spain; laepoi@ibmcp.upv.es (L.P.); bbelda@ibmcp.upv.es (B.B.-P.); 2Biotechnology Application and Research Center, and Department of Molecular Biology, Faculty of Science and Letter, Bilecik Seyh Edebali University, 11230 Bilecik, Turkey; dilek.unal@bilecik.edu.tr

**Keywords:** polyamines, spermidine, thermospermine, nonsense-mediated decay, no-go decay, non-stop decay, quality control, translation

## Abstract

Plant polyamines (PAs) have been assigned a large number of physiological functions with unknown molecular mechanisms in many cases. Among the most abundant and studied polyamines, two of them, namely spermidine (Spd) and thermospermine (Tspm), share some molecular functions related to quality control pathways for tightly regulated mRNAs at the level of translation. In this review, we focus on the roles of Tspm and Spd to facilitate the translation of mRNAs containing upstream ORFs (uORFs), premature stop codons, and ribosome stalling sequences that may block translation, thus preventing their degradation by quality control mechanisms such as the nonsense-mediated decay pathway and possible interactions with other mRNA quality surveillance pathways.

## 1. Introduction

The eukaryotic cell has developed sophisticated mechanisms to cope with alterations that may occur during the complex process of gene expression, which could lead to detrimental consequences. Until the final gene product is released as a functional protein, numerous sequential steps such as transcription, mRNA maturation, export from the nucleus, translation, and folding demand checkpoint controls to avoid the production of defective proteins that could compromise cell viability. The translational process is usually monitored by ribosome-associated quality control pathways able to recognize faulty events caused by halted or stalled ribosomes during the process of mRNA translation [1]. These surveillance pathways are recruited to dismantle stalled translational complexes and to degrade or recycle mRNAs, ribosomes, and aberrant nascent polypeptides to complete the clearing process of defective components and products of the translation machinery. Some of these pathways address the removal of defective polypeptides resulting from improper folding, while other specialized pathways become activated upon ribosome stalling due to defects in mRNA, translational components, or the nascent polypeptides [2]. In addition to other post-translational regulatory mechanisms, these supervision pathways participate in the course of proteostasis or protein homeostasis in the cell. 

As the ribosome becomes the hub for the activity of the quality control pathways, any of the cellular metabolites or proteins linked to the ribosomal activity may impinge on the activity of the clearing pathways. In this review, we focus on the role of PAs as metabolites derived from amino acid catabolism universally present in eukaryotic cells, with essential functions linked to ribosomal activity. In particular, we compile recent information of two PAs, Tspm and Spd, whose activities in translation preventing the occurrence of ribosome stalling qualify them as quality keepers, indirectly protecting mRNAs from their degradation by preventing the activation of ribosome-associated mRNA quality control pathways. 

## 2. Polyamines and Interactions with the Translation Machinery

PAs are small aliphatic compounds positively charged at physiological pH and are widely present in nature. PAs are enzymatically produced by the decarboxylation of the amino acids ornithine and/or arginine to provide the precursor putrescine and by the sequential addition of aminopropyl groups from decarboxylated *S*-adenosylmethionine (dcSAM) to yield higher-molecular-weight polyamines (Figure 1). The most abundant PAs in eukaryotes are the diamine putrescine (Put), the triamine Spd, and the tetraamine spermine (Spm). Plants and algae display diversity in PA biosynthesis [3], and, like some microbial thermophiles, they also contain a Spm unsymmetrical isomer named Tspm involved in plant vascular development [4]. 

The chemical nature of PAs has driven studies on their binding properties with respect to negatively charged macromolecules such as DNA, RNA, and phospholipids. Pioneering studies on the in vivo distribution of two highly abundant PAs (Spd and Spm) bound to macromolecules revealed that most of these two PAs in rat liver remained as a complex with RNA and to a minor extent was bound to DNA and phospholipid [5]. Previous studies had already shown that PAs could stimulate translation efficiency and fidelity in prokaryotic and eukaryotic cell-free systems, and the enhanced fidelity occurred at the level of aminoacyl-tRNA binding to the ribosome [6,7]. Moreover, by stimulating the methylation of adenine residues close to the 3’-end of the 16S rRNA, PAs were shown to be required for the efficient assembly of 30S ribosomal subunits in *E. coli* [8,9]. Refined kinetic studies with fluorescence-based assays have shown that PAs promote codon recognition by stimulating the binding of the ternary complex to the ribosome [10], in agreement with structural information obtained with photoreactive PA analogues [11,12,13]. Altogether, these pioneering studies strongly promote the idea that PAs are involved in translation through interactions with RNA and ribosomes [14]. In addition to their direct impact as general stimulators of translation, PAs have been shown to fine-tune the translation of specific highly regulated mRNA coding sequences by enhancing ribosomal frameshifting [15]. 

## 3. Polyamines and mRNA Quality Control Mechanisms

Recent information on the function of two plant PAs, Tspm and Spd, has uncovered unexpected roles for PAs related to faulty translational events. These novel functions for PAs related to ribosomal activity are connected to specialized mRNA surveillance mechanisms that can detect mRNAs with defective translation and initiate their selective degradation. There are at least three known mRNA anomalies that lead to activation of the specific quality control pathways. One abundant alteration among mRNA sequences is the presence of premature termination codons (PTCs) that may lead to the production of defective proteins with dominant negative effects, in this case the surveillance machinery engaged is the nonsense-mediated decay pathway (NMD). Another type of activated system is the no-go decay pathway (NGD) that acts when ribosomes halt during elongation due to the presence of secondary mRNA structure, of rare codon repeats, or because of structural features of the nascent polypeptide that block elongation. A third route that takes care of translational failures is the non-stop decay pathway (NSD), operating on mRNAs lacking natural stop codons. In addition to the surveillance and degradation of mRNAs halted during translation, specialized complexes take care of the removal of non-functional proteins, a process described in detail elsewhere [16]. In the next subsections and sections, we briefly update each of these mRNA surveillance systems and the links to PAs respectively. 

### 3.1. The NMD Pathway

The NMD pathway targets mRNAs carrying PTCs whose presence may generate truncated dominant-negative proteins with deleterious consequences for the cell [17,18,19]. The occurrence of PTCs can have multiple origins, either by non-productive alterations such as random DNA mutations or programmed genomic rearrangements, or by endogenous errors during transcription or RNA splicing [20]. PTCs can also happen in natural mRNAs as in those with upstream open reading frames (uORFs) and in transcripts of pseudogenes, transposons, and even non-coding RNAs [21,22,23]. Recent studies on *A. thaliana* revealed that around 1% of coding mRNAs and up to 20% of mRNA-like non-coding RNAs are targets of the NMD pathway [23]. Moreover, among the mRNA targets, a remarkable 17.4% of the multi-exon coding genes with splicing variants have been allocated to the NMD pathway, thus highlighting the importance of the alternative splicing coupled to NMD (AS-NMD) regulation [24,25]. The most important *trans*-acting factors involved in the NMD, named up-frameshift proteins (UPFs), were initially described in *S. cerevisiae* [26], and they have been shown to be highly conserved among eukaryotes [27] with only minor mechanistic differences for plants [28]. *A. thaliana upf* mutants defective in the NMD pathway have uncovered important functions for this pathway in development and stress responses [29,30,31,32]. In addition to the UPF proteins, accessory factors participate in the sequential steps of PTC detection, tagging, and destruction by endo- and exo-ribonucleases (exosome and Xnr1), all of them studied and described elsewhere with great detail [33,34]. Although the comprehensive mechanisms of NMD target recognition are still a matter of debate, there is a consensus on the participation of canonical translation termination factors eRF1 and eRF3 in the NMD process, revealing a new unexpected hypothesis as to whether this pathway not only watches for the presence of PTCs but also acts as a quality-control pathway that continuously identifies errors of premature termination during the translation process [35]. 

### 3.2. The NGD Pathway

This surveillance pathway detects ribosomes that have been stalled during translation elongation because of specific features of nascent polypeptides, secondary mRNA structures that block ribosome progression, or rare codon repeats that lead to prolonged vacancy of the A-site of the ribosome. It counts with specialized protein factors described initially in yeast and named Dom34 and Hsb1 (Dom34:Hsb1) and later in mammals and known as Pelota and HSb1-like (PELO:HSB1L) [36]. Both proteins are highly conserved and present also in plants [37,38]. The complex Dom34:Hbs1 is structurally similar to termination factors eRF1:eRF3 and in fact they do bind the vacant A-site of the ribosome to promote dissociation of the aberrant elongation complex, induce mRNA endonucleolytic cleavage, and facilitate ribosome recycling [39]. After the cleavage, the resulting mRNA fragments are subsequently degraded by the exosome and Xnr1 exonucleases like in the NMD pathway. Several questions remain to be elucidated for this pathway. First, mRNA cleavage can occur in the absence of Dom34:Hbs1 and occurs upstream of the stalling site [40]. Second, no elongation block leads to activation of the NGD, since (i) cycloheximide treatment stabilizes mRNAs and inhibits their decapping [41] and (ii) in other cases ribosomal pauses provide biological functions [42]. Therefore, the cell machinery must distinguish between useful ribosome elongation pauses or unwanted ribosomal arrests that require activation of the NGD. In this sense the degradation machinery may perceive whether the A-site is unoccupied or not and for how long this aberrant situation persists since other pathways such as translational frameshifting may be favored. An attractive model based on recent data suggests that sensing ribosome collisions trigger the endonucleolytic cleavage upstream of the stall site [43], though the identity of such endonuclease is still unknown.

### 3.3. The NSD Pathway

The NSD pathway degrades mRNAs that lack in-frame stop codons, either because of premature polyadenylation leading to non-stop mRNAs with poly(A) or by endonucleolytic cleavage within the open reading frame producing non-stop mRNAs lacking poly(A) [40,44]. In the case of poly(A) translation, the presence of encoded poly-lysine causes ribosome stalling probably due to interactions with negatively charged rRNA in the exit tunnel, although the nature of the poly(A) sequence also leads to a ribosome sliding behavior [45,46]. The activity of this pathway was first shown in yeast and mammals [47,48], and it has been recently shown to be operative in higher plants [49]. This pathway shares components with the NGD since a key player that was found in yeast (Ski7), structurally related to Hsb1 and eRF3, is not present in other eukaryotes. Therefore, organisms lacking Ski7 rely on Dom34/Hbs1 proteins that work for both the NGD and the NSD [40].

## 4. Thermospermine and uORF-Dependent Translational Inhibition: Is There A Connection to the NMD Pathway?

Tspm was first discovered in thermophilic bacteria [50], and it was later shown to be widely distributed in the plant kingdom but absent in animals and fungi [3]. Tspm, a structural isomer of Spm, is converted from Spd by the activity of the enzyme thermospermine synthase (ACL5) identified in *A. thaliana* after the isolation of a mutant deficient in Tspm named *acaulis 5* (*acl5*) [51]. The first description misidentified the enzyme encoded by *ACL5* as a spermine synthase since the analytical procedures used were unable to distinguish between Spm and Tspm [52], but it was later shown that ACL5-like enzymes synthesize Tspm both in diatoms and *A. thaliana* [53]. Different from other aminopropyltransferases, the ACL5 enzyme was probably acquired by a plant ancestor by means of horizontal transfer from archaea or bacteria [54]. As an example of how PAs, as plant growth regulators, are linked to other hormonal signals, Tspm biosynthesis has been shown to be regulated by the negative feedback regulation of an auxin signaling module [55,56]. The role of Tspm in plant development is highly specialized in vascular development with key functions in xylogenesis by preventing premature cell death [57,58,59]. Precisely the dramatic phenotype of the *acl5* mutant displaying stunted growth allowed for the genetic isolation of mutant suppressors, named *sac*, able to restore stem growth in the absence of Tspm to illuminate on its molecular functions [60,61,62]. Interestingly, several of the isolated suppressors mapped to ribosomal proteins or ribosome-associated proteins, thus suggesting molecular functions for Tspm related to the regulation of translation. Indeed, among the identified *SAC* genes, *SAC51* and its family of bHLH-like transcription factors (*SACL1-3* genes) have been key to elucidating downstream functions of Tspm in higher plants [63]. Although molecular details for the mechanism of action of Tspm still remain unknown, its downstream target functions through the activity of SACL proteins have been shown to rely on the interplay between heterodimers of bHLH transcription factors involved in vasculature formation [64]. 

One common feature of *SACL* genes is the presence of uORF in their 5’-leader sequence, with the peculiarity that several suppressor mutations map precisely in this region [60]. The isolation of the *sacl* mutants that no longer require Tspm indicates that Tspm is required to bypass the uORF-dependent translational inhibition of the *SACL* main ORF (mORF). The presence of uORFs, in particular those encoding conserved peptide sequences (CPuORFs), as is the case of the *SACL* genes, is often a feature to define substrates of the NMD pathway [18,23]. The genetic studies with *upf* mutants seem to corroborate this idea, as gene expression studies have shown the upregulation of the *SACL* genes in these mutants, suggesting that UPF proteins are required for the suppression of the *SACL* mRNAs [29,30,31,32]. It is therefore conceivable that Tspm activity in *SACL* translation helps to evade the premature termination on their uORFs, thus protecting these mRNAs from degradation by the NMD pathway (Figure 2). The precise mechanism of Tspm action to favor the translation of the mORF [65] is still an open question and whether this polyamine directly works within the ribosome or as an external factor is an active line of research nowadays. Another intriguing question to be solved is whether any other mRNA containing uORF could be considered as a Tspm client, as more than 20,000 uORFs have been identified in *A. thaliana* [66], although the restricted tissue expression pattern of ACL5 would certainly limit the number of potential Tspm targets containing uORFs. 

## 5. Spermidine and Hypusination of eIF5A: How to Avoid Reversible Ribosome Pausing and Links to the NMD Pathway

The triamine Spd is a very ancient PA, and its biosynthetic enzyme spermidine synthase (SPDS) has been catalogued as one of the few protein families potentially present in the last-universal-common-ancestor (LUCA) [67]. More empirical studies have shown that Spd is an essential metabolite required for cell growth and survival in eukaryotic organisms [68,69,70]. One role of Spd that can explain its essential nature is its requirement as a donor for the activation by post-translational modification of the translation factor eIF5A [71]. This translation factor is the unique protein known to undergo the enzymatic modification of a conserved lysine to the unusual amino acid hypusine (Nε-(4-amino-2-hydroxybutyl)lysine) [72] by the sequential activity of first the deoxyhypusine synthase (DHS), forming the reversible intermediate deoxyhypusine, and second the deoxyhypusine hydroxylase (DOHH), catalyzing irreversibly the required hydroxylation to yield the hypusine residue [73]. As in the case of the Spd, the hypusination of translation factor eIF5A has been shown to be essential for eukaryotes’ cell viability [74,75,76,77]. The eIF5A nomenclature and its assigned functions on translation have suffered chronological alterations, as it was initially named IF-M2Bα, later eIF4D, and finally eIF5A after its description as a protein that is able to stimulate the formation of methionyl-puromycin in vitro [78,79,80]. Hypusine discovery [81] preceded its characterization as an activator of the translation factor eIF5A and its biosynthetic origin from Spd by several years [82], but it was not until several decades later that eIF5A was shown to fulfill translation elongation rather than initiation functions in yeast [83]. To provide more complexity to the eIF5A functions carried out within the ribosome, recent studies in yeast and humans have shown that eIF5A is not only involved in elongation but also in the translation termination process [84]. Both eIF5A and DHS are present and essential in eukaryotes and archaea (named aIF5A) [85], but they are absent in eubacteria [86]. However, eubacteria possess a structurally related elongation factor named EFP that also suffers analogously though biochemically unrelated modifications essential for EFP activity [87,88,89]. As a difference to aIF5A and eIF5A, the knock-out of EFP is viable for the bacteria, though the mutants are more sensitive to antibiotics [90]. Structural analyses have located both EFP and eIF5A within the ribosome between the P- and the E-sites, thus supporting their role as translation elongation factors [91,92,93]. A major breakthrough in the field of eIF5A/EFP was the elucidation of their roles as factors required for the translation of mRNAs that encode proline rich-repeat proteins that, in the absence of these factors, lead to ribosome stalling [94,95,96], caused by the problematic chemical structure of proline as a rigid secondary amine being a poor peptidyl donor and receptor [97]. It is remarkable that bioinformatics analysis of poly-proline rich proteins in different eukaryotic systems from yeast to plants has suggested the idea of a parallel specialization of the Spd-eIF5A pathway with the functional organization of proline rich-repeat proteins [98]. This information has facilitated the allocation of molecular mechanisms to Spd by means of its role on the eIF5A hypusination, such as its role in yeast fertility by promoting the translation of formin, a poly-proline rich containing protein, involved in actin cytoskeleton remodeling during yeast mating [99]. The hypusination pathway in higher plants has been involved in the response to both biotic and abiotic stress conditions as well as in plant developmental processes [98,100,101,102,103] though the potential targets at the level of mRNA translation remain to be identified. 

The way eIF5A/EFP resolves problematic poly-proline ribosome pauses is among the topics of excellent reviews [104,105] that summarize recent studies on this subject. One key aspect of the strategy used by eIF5A/EFP to alleviate ribosome pauses is related to their location within the ribosome, as they enter the ribosome through the E-site probably as a signal of slow translation elongation or termination that demands their intervention as a case of reversible pause. The structural studies favor the model that hypusine stabilizes the acceptor arm of the P-site peptidyl-tRNA at the peptidyl transferase center rather than contributing to the catalysis of the peptide-bond formation. This is in contrast with vacant A-sites generated by non-optimal codons, truncated mRNA, or nascent polypeptide features that, when sensed as an irreversible arrest, can trigger the degradation of the mRNA upon activation of the NGD pathway. As mentioned before, the ribosome-associated cell machinery uses advanced biochemical sensing systems to distinguish between a transient stall and a dead arrest that demands either reversible action or destructive/recycling intervention, respectively. 

Although no evidence has been reported to date for the interaction of eIF5A with the NGD and NSD pathways, many studies have linked the functions of eIF5A to the NMD pathway. Pioneering searches for yeast mutants with increased stability of mRNA targets of the NMD pathway have uncovered eIF5A as an essential protein involved in mRNA decay [106]. This observation was later confirmed with double genetic disruption of yeast eIF5A-encoding genes complemented with temperature-sensitive human *EIF5A1* gene, showing elongated half-lives of NMD transcripts [107]. More recently, the functional connection between eIF5A activity and the NMD pathway has been extended to human cells [108]. One possible explanation for the increased stability of NMD-related targets is the unexplained functional link of eIF5A with stress-granule assembly [109]. Altogether these studies support the idea that Spd-dependent hypusination of eIF5A provides an NMD-related proofreading activity in translation although the underlying biochemical basis remains to be elucidated (Figure 3). 

## 6. Conclusions

PAs are well-conserved metabolites with essential functions in eukaryotes. Among the different roles and locations within the cell, a considerable number of studies have linked their activities to the ribosome. This highlights vital functions for PAs, such as the essential nature of Spd for eukaryotes and the absolute requirement of Tspm for plant vascular development. The mode of action for Spd within the ribosome is to provide the aminobutyl moiety to activate the translation factor eIF5A, whereas for Tspm the available information suggests a direct and probably intimate role with the ribosome machinery. In both cases, a reduction in their levels correlates with stabilization of mRNAs belonging to the NMD pathway, thus indicating that both PAs may act as watchers for potential ribosome conflicts. In the case of Tspm, the known NMD targets belong to the group of genes with uORF sequences containing PTCs that act as translation inhibitors for the mORF. Unfortunately, the mechanistic molecular details of Tspm activity in the ribosome still need to be deciphered. On the other hand, the suppression of Spd-mediated hypusination of eIF5A leads to an accumulation of unstable transcripts, including some of the AS-NMD type, thus suggesting unknown connections of the eIF5A translation factor with the splicing machinery. Still, how hypusinated eIF5A connects to the NMD pathway awaits further studies. We cannot forget that the inactivation of eIF5A could lead to the activation of either NGD or the NSD pathways. For instance, if poly-proline induced ribosomal blocks are not rapidly resolved and lead to ribosomal collisions, this could activate NGD machinery. On the other hand, alterations in the termination process by inactive eIF5A may lead to non-stop ribosomal activity, which could activate the NSD pathway. What has been well documented is that both Tspm and Spd (through the activity of eIF5A) seem to act as molecular sentinels to prevent translation anomalies that may lead to activation of the NMD machinery and subsequent mRNA degradation. In this review, we have focused on two PAs that share similar proofreading functions on translation, but it would not be a surprise to find out that other PAs may fulfill related functions on this or related cellular processes. We anticipate that in the near future the roles of PAs in the cell in relation to ribosomal activity will be fully elucidated, thus helping to solve one of the remaining mysteries of molecular biology [110]. 

## Figures and Tables

**Figure 1 plants-08-00109-f001:**
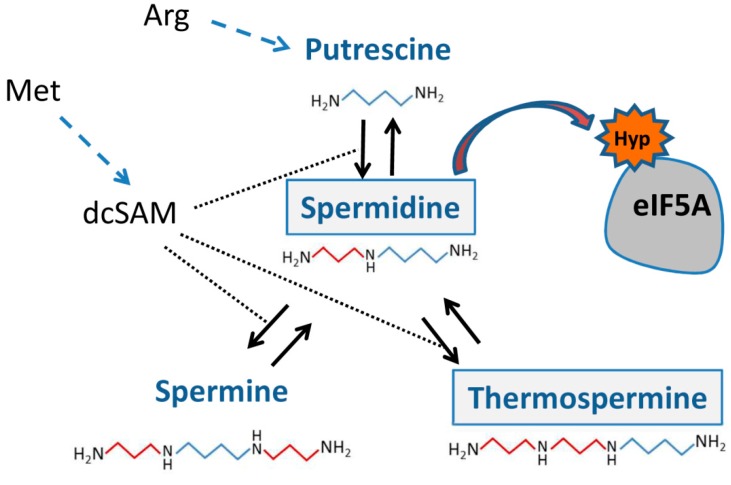
Polyamines derive from amino acid catabolism. In plants, the prevalent biosynthetic pathway uses arginine as a precursor for the synthesis of the diamine Put to which enzymatic addition of aminopropyl groups (in red) from decarboxylated SAM (dcSAM) yield the triamine Spd and subsequently the tetraamines Spm and Tspm. The post-translational transfer of the aminobutyl moiety (in blue) from Spd to a conserved lysine of eIF5A leads to the modified hypusine residue (Hyp) that renders an active translation factor.

**Figure 2 plants-08-00109-f002:**
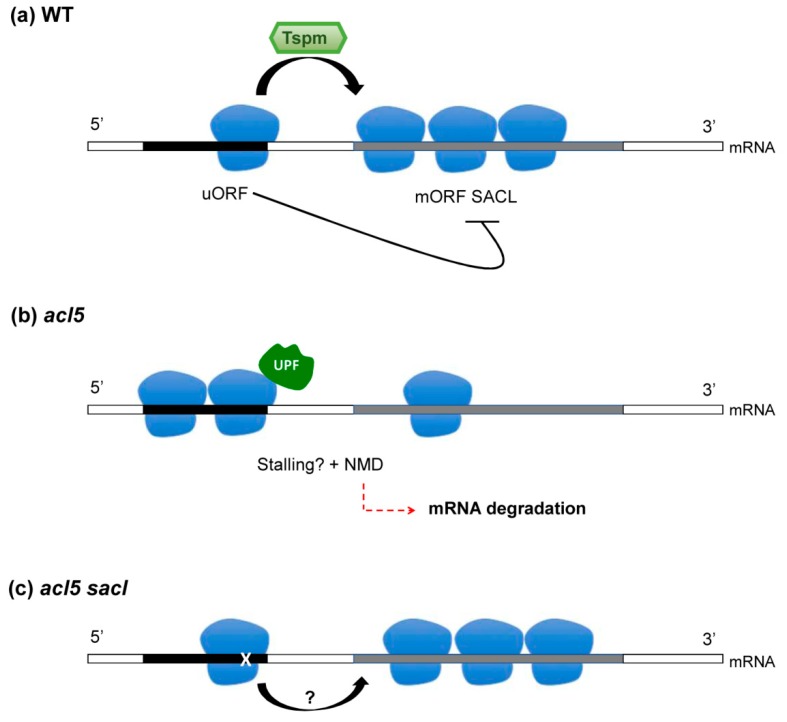
Thermospermine action on the translation of *SACL* genes and the connection to the NMD pathway. Under normal conditions, (**a**) the presence of Tspm, by unknown mechanistic details, prevents the blockade of translation of the main ORF (in grey) of *SACL* genes imposed by the presence of inhibitory upstream ORF (in black). *SACL* genes belong to the NMD substrate category, so it can be envisaged (**b**) that in the *acl5* mutant lacking Tspm the ribosomes may stall at the uORF and activate the NMD pathway. In the double mutants *acl5 sacl* (**c**), the presence of *cis* mutations (X in white) in the uORF of *SACLs* may alleviate ribosome stalling by changes in the peptide sequence to allow translation of the mORF or by other mechanisms to be elucidated.

**Figure 3 plants-08-00109-f003:**
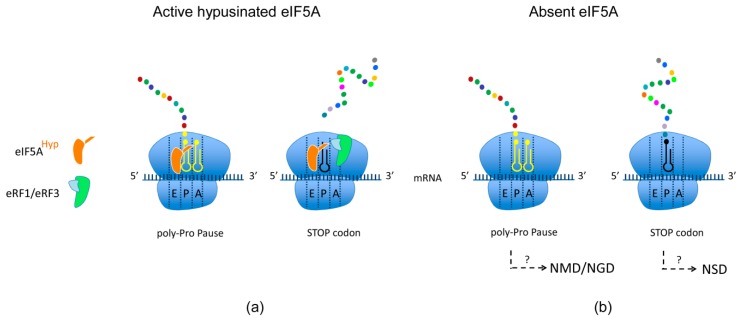
Spermidine-mediated hypusination of eIF5A and links to surveillance pathways. (**a**) Hypusinated eIF5A promotes elongation of stalling motives like poly-proline and stimulates eRF1-mediated peptidyl hydrolysis. (**b**) Non-functional eIF5A generates a vacant E-site during ribosome stalling at poly-proline elongation and affects termination, leading to the accumulation of NMD substrates by unknown mechanisms. Absence of eIF5A could also potentially affect NGD and NSD pathways during elongation and termination.

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
