# Peer review of "Polyamines as Quality Control Metabolites Operating at the Post-Transcriptional Level"

_plants, 2019, doi:10.3390/plants8040109_

Reviewer 1 Report

I have read the manuscript entitled, “Polyamines as quality control metabolites operating at the post-transcriptional level”.

This is an interesting study describing the role of two polyamines, spermidine and thermospermine, in some molecular functions related to quality control pathways for regulated mRNAs at the translational level.

The title of the review is "Polyamines..." but only describe the role of Tspm and Spd in detail. The role of Tspm is very speculative but I think that is an interesting hypothesis. The review would be improved if the role of other polyamines at the translational level was described, even though it was done in less detail than Tspm and Spd.

Author Response

The reviewer is right that the text related to the roles of polyamines in relation to ribosomes and translation independent on the quality control pathways, was not properly highlighted. Although we had introduced the topic at the beginning of section 2, it was not properly separated to uncover its importance. Therefore we have split the original section 2 into two sections. Now section 2 refers only to general interactions between polyamines and translation, whereas new section 3 focuses on the interactions with the quality control pathways. We hope that now the review is better balanced on the issue raised by the reviewer. 

Reviewer 2 Report

The review on "Polyamines as quality control metabolites operating at the post-transcriptional level" by L. Poivedin  et al., report on the plant polyamines's roles and more specifically, spermidine (Spd) and thermospermine (Tspm) on regulating post-transcriptional mechanisms. Both of them share molecular functions related to quality control pathways for tightly regulation of the the mRNAs at the level of translation. The roles of Tspm and Spd and the mechanisms by which they facilitate translation of mRNAs containing upstream ORFs (uORFs), premature stop codons and ribosome stalling sequences are nicely presented and thoroughly reviewed. Both these polyamines share similar proofreading functions on translation through quality control mechanisms such as the nonsense-mediated decay pathway and possible interactions with other mRNA quality surveillance pathways. The review contains useful information on the latest findings of possible functions and activities of these PAs on the ribosome and its content would be of great interest for the researchers in the field.

        One suggestion: It would be very useful to add a figure with the relevant biosynthesis pathway of these two polyamines in the text.

Author Response

We have followed the reviewer suggestion to introduce a new figure 1 to schematically present the PA biosynthetic pathway highlighting the presence of Spd and Tspm.

Reviewer 3 Report

The present MS is a review on the role of polyamines in translation.  There are a lot of studies about the role of poilamines in stress tolerance, their direct protective effects, and also their possible role in stress signalling, but there are only a few notifications and references about their relation with mRNA quality control. 

The present MS is well-written, the grammer is good, the text and figures understandable.  

This review shows a new direction for research interest.

 I would like to suggest some additional references: 

1. Roles of polyamines in translation Dever and Ivanov (2018) 

2. Hypusine, a polyamine-derived amino acid critical for eukaryotic translation*  Park and Wolff (2018)

3. Thermospermine enhances translation of SAC51 and SACL1 in Arabidopsis. Yamamot and Takahashi (2017)

Author Response

As suggested by the reviewer, we have included the references indicated in the appropriate sections.

Reviewer 4 Report

I think the manuscript well written, clear and concise. The title conveys the main findings of the study. The abstract adequately summarizes the manuscript. All subchapters are well written. Authors explain the wider implications of the review (e.g. Tspm; Spd). Figures are mostly well designed and effectively illustrate the most important knowledge. References include the most relevant to the subject.

I would like to suggest just some minor correction:

Please insert references to the first paragraph of the introduction (L. 32; L. 39).

I suggest some summarized sentences about the selective degradation of non-functional proteins (L. 74-89).

Please add more accurate description all of the abbreviations to the figure legends. Please add “mRNA” inscription to all lines in Fig 1. and 2.. I suggest a “?” or more accurate notation to Fig.1. c. Arrow (direction of translation) is missing in Fig 2. Please use other colour on fig 2, which can be better in black and white printed form.

Author Response

We have performed the following changes according to useful reviewer suggestions: (i) we have added two new references to the first paragraph of the introduction, (ii) we have  modified new Fig.2 by adding mRNA inscription and question mark symbol in new Fig.2c to indicate unknown mechanisms, (iii) we have modified new Fig.3 to make visible any black & white printing and addition of 5’-3’ symbols and mRNA to indicate direction of translation. Concerning the selective degradation of non-functional proteins (out of the scope of this review), we include cite 13.

Round  2

Reviewer 1 Report

Authors have improved  the manuscript. I accept it in present form for its publication